# Combined Neuromuscular Electrical Stimulation and Elastic Taping Improves Ankle Range of Motion Equivalent to Static Stretching in Untrained Subjects

**DOI:** 10.3390/jfmk10010058

**Published:** 2025-02-06

**Authors:** Riyaka Ito, Tatsuya Igawa, Ryunosuke Urata, Shomaru Ito, Kosuke Suzuki, Hiroto Takahashi, Mika Toda, Mio Fujita, Akira Kubo

**Affiliations:** 1Department of Physical Therapy, Graduate School of International University of Health and Welfare, 2600-1 Kitakanemaru, Ohtawara 324-8501, Tochigi, Japan; riyakaito@gmail.com (R.I.); hiro9taka6@gmail.com (H.T.); todamika2@gmail.com (M.T.); miofujita24@gmail.com (M.F.); akubo@iuhw.ac.jp (A.K.); 2Department of Rehabilitation, International University of Health and Welfare Hospital, 573-3 Iguchi, Nasushiobara 329-2763, Tochigi, Japan; 3Department of Physical Therapy, School of Health Sciences, International University of Health and Welfare, 2600-1 Kitakanemaru, Ohtawara 324-8501, Tochigi, Japan; 4Innovative-Rehabilitation Center, New Spine Clinic Tokyo, 2-6-3 Hirakawacho, Chiyoda-ku, Tokyo 102-0093, Japan; ryu.urata62@gmail.com; 5Department of Physical Therapy, School of Health Sciences at Odawara, International University of Health and Welfare, 1-6 Minamicho, Odawara 250-0013, Kanagawa, Japan; shomaru.ito5151@gmail.com; 6Department of Rehabilitation, Yamagata Saisei Hospital, 79-1 Okimachi, Yamagata 990-0818, Yamagata, Japan; kousuke.s0922@gmail.com

**Keywords:** ankle joint range of motion, neuromuscular electrical stimulation, elastic tape, equivalence test

## Abstract

**Background/Objective**: Maintaining sufficient ankle joint range of motion (ROM) contributes to efficient movement in sports and daily activities. Static stretching (SS), while effective, demands significant time, highlighting the need for alternative, time-efficient approaches to improve ROM. Therefore, this study aimed to evaluate the effectiveness of combined intervention (CI) using neuromuscular electrical stimulation (NMES) and elastic tape versus SS. **Methods**: This randomized crossover trial was conducted in healthy university students. They underwent both interventions with a 1-week washout period. The CI entailed the application of elastic tape to the plantar surface of the foot coupled with NMES targeting the posterior lower leg muscles for 1 min. SS was administered for 5 min using a tilt table. Outcome measures included the dorsiflexion angle (DFA), finger-floor distance (FFD), straight leg raise (SLR) angle, plantar flexor strength (PFS), and knee flexor strength (KFS), assessed pre- and post-intervention. DFA was analyzed using equivalence testing with a predefined margin. **Results**: Both interventions yielded significant improvements in DFA, FFD, and SLR. The combination of NMES and elastic tape demonstrated equivalence to 5 min of SS in enhancing DFA. Neither intervention resulted in a significant reduction in PFS or KFS. **Conclusions**: The CI of NMES and elastic tape effectively and safely improves flexibility in a short time. Its time efficiency makes it a promising alternative to SS, especially for brief warm-ups or limited rehabilitation time. Further research should explore its long-term effects and broader applicability.

## 1. Introduction

Reduced ankle joint range of motion (ROM) is a well-established risk factor for lower limb injuries and functional impairments. Additionally, limited ankle ROM is associated with an increased risk of ankle sprains, Achilles tendinitis, and anterior cruciate ligament injuries [1,2]. It also negatively affects one’s standing balance ability [3] and restricts athletic performance. Therefore, ankle ROM is a crucial parameter for both daily activities and athletic performance. For athletes and older adults, in particular, maintaining or improving joint mobility is imperative for preventing injuries and ensuring functional independence.

Traditional interventions to improve ROM include static stretching (SS), dynamic stretching, massage, and resistance training. Among these, SS is widely regarded as the standard method, with meta-analyses demonstrating significant improvements in ankle ROM following interventions lasting between 5 and 15 min [4]. However, the time-intensive nature of SS presents practical challenges, particularly in pre-competition warm-ups and time-constrained daily exercise routines. Moreover, prolonged SS may induce temporary reduction in muscle strength [5,6], raising concerns regarding its potential impact on physical performance.

Recently, alternative approaches such as neuromuscular electrical stimulation (NMES) and elastic tape applications have gained attention. NMES, which delivers external electrical stimulation to induce muscle contraction, is widely employed to enhance muscle strength and facilitate muscle re-education [7]. Additionally, NMES alleviates pain and elevates pain thresholds [8,9], indicating its potential role in improving ROM. Pain alleviation may improve ROM by reducing muscle tension that restricts joint mobility and by decreasing movement-avoidance behaviors. Elastic tape, known for its effects on fascial dynamics, has emerged as a simple and practical intervention for enhancing ROM [10,11]. Notably, applying elastic tape to the sole of the foot improves ankle ROM via fascial mechanisms, making it particularly suitable for short-term application before athletic performance. Furthermore, the application of elastic tape exerts shear forces that influence joint movement [12], and it has also been reported to reduce muscle stiffness and pain [13,14].

Our previous research demonstrated that a combination of NMES and SS effectively improved ROM within a brief intervention period of 4 min [15]. Furthermore, the application of elastic tape to the foot may yield effects comparable to those of 30 s of SS [10]. These findings highlight the potential of short-duration interventions to overcome the limitations of traditional SS and offer practical solutions for athletic warm-ups and flexibility enhancement in everyday life.

This study aimed to evaluate the effectiveness of a combined NMES and elastic tape intervention in improving ankle ROM more efficiently than SS alone. It also aimed to establish the non-inferiority and equivalence of the combined approach. The results of this investigation are expected to advance innovative therapeutic strategies for enhancing ankle ROM with broad applications in both sports and daily life.

## 2. Materials and Methods

### 2.1. Ethics Statements

The study protocol was approved by the institutional review board of the research institution and adhered to the principles outlined in the Declaration of Helsinki (approval ID: #24-TC-002, date: 21 August 2024). Comprehensive written and verbal explanations of the study were provided to all participants, and informed consent was obtained prior to participation. The trial was prospectively registered in the University Hospital Medical Information Network Clinical Trials Registry under identifier UMIN000054631.

### 2.2. Study Design and Population

This randomized crossover trial used a non-inferiority and equivalence design to compare the effects of a combined intervention (CI) involving NMES and elastic tape with those of SS on ankle dorsiflexion angle (DFA) in healthy university students. Healthy university students were recruited as participants through university bulletin board announcements. The required sample size was determined using G*Power software (version 3.1.9.4; Heinrich-Heine-Universität, Düsseldorf, Germany), based on an effect size of 0.44 [16], an α level of 0.05, and a statistical power of 0.8, yielding a sample size of 83 participants.

The inclusion criteria were physically active university students who exercised at least once per week. Exclusion criteria included (1) lower limb injuries within the past 6 weeks, (2) lumbar spine conditions within the past 6 weeks (e.g., diagnosed lumbar lesions restricting ROM, prior lumbar surgery, or impairments of the lumbosacral spine affecting ROM or function); and (3) a history of lower limb surgeries within the past 6 months or major ligament surgeries within the past year.

### 2.3. Randomization

Randomization was conducted using GraphPad (version Prism 10.2.3; GraphPad Software Inc., San Diego, CA, USA), with the allocation table generated by a researcher blinded to the participant characteristics. A 1:1 block randomization algorithm stratified by sex was used to ensure balanced group assignments. The participants were randomized into either Group A or Group B. Allocation concealment was maintained through a central registration system, and an independent researcher uninvolved in the creation of the allocation table assigned participants to their respective groups sequentially based on the order of registration.

### 2.4. Intervention

The participants in Group A received CI, which consisted of NMES applied to the triceps surae muscles in conjunction with elastic tape applied to the plantar surface of the foot. The participants in Group B underwent SS targeting the triceps surae muscles. A 1-week washout period was implemented to minimize any residual effects of the initial intervention. Following this period, participants in both groups were crossed over to receive alternative interventions.

#### 2.4.1. CI

Elastic tape (Iotape 38 mm; RINDSPORTS, Osaka, Japan) was prepared by cutting it to 50% of the length of the participant’s foot and was applied to the plantar surface of the dominant foot. The tape was positioned 3 cm posterior to the heel, with the initial 1 cm serving as an anchor without tension and the remaining segment stretched to 120% of its original length. NMES was subsequently applied to the posterior lower leg muscles for 1 min. For the NMES application, self-adhesive electrodes (5 × 9 cm) were positioned over the motor points of both the medial and lateral gastrocnemius muscles to ensure effective stimulation. A low-frequency stimulator (ESPURGE; ITO, Saitama, Japan) was used, configured with a pulse width of 250 µs, a frequency of 80 Hz, and a stimulation cycle consisting of 3 s of contraction followed by 3 s of rest. The stimulation intensity was adjusted to the maximum level that the participant could tolerate, with encouragement to increase the intensity to the upper limit, ranging from 20 mA to 50 mA.

#### 2.4.2. SS

SS of the posterior lower leg muscles was performed using a tilt table equipped with a fixed backrest and an adjustable footplate. Each participant stood with the knee extended, and the footplate angle was set to the maximum angle they could tolerate without excessive pain. This angle was then maintained throughout the 5-min stretching period, ensuring a consistent intensity of stretch for that individual. Although the absolute angle varied among participants, fixing the footplate for each participant minimized within-participant variability. By standardizing the backrest and footplate setup, we were able to provide a stable, uniform stretching protocol that enhanced reproducibility and controlled for differences in stretching intensity over time.

### 2.5. Outcome Measures

The primary outcome measures were DFA, finger-floor distance (FFD), and straight leg raise (SLR) angle. SLR was included to assess overall changes in ROM throughout the body, as the intervention might influence the broader fascial and musculoskeletal systems. DFA was assessed using a digital inclinometer (Apple Inc., Cupertino, CA, USA). The participants were instructed to tilt their lower leg forward without lifting their foot off the floor, and the evaluator measured DFA using a smartphone fixed at the center of the ventral surface of the participant’s lower leg. This method, as previously reported by our group, demonstrates high inter-rater reliability [11]. DFA measured using a smartphone corresponded to the maximum forward tilt angle of the lower leg with the floor surface set to 0°. Consequently, smaller DFA values reflect a greater ankle dorsiflexion ROM. FFD was measured using a digital flexibility testing device (T.K.K.5403; Takei Scientific Instruments Co., Ltd., Saga, Japan). SLR was assessed using the same digital inclinometer smartphone application used for the DFA measurement. The participants were positioned supine, and the smartphone was fixed along the midline of the lower leg. The inclinometer was calibrated to 0° before the lower limb was raised, and the maximum leg-raise angle was recorded.

Plantar flexor strength (PFS) and knee flexor strength (KFS) were evaluated as secondary outcomes using a handheld dynamometer (Mobie; Sakai Medical Co., Ltd., Osaka, Japan) to ensure that none of the interventions adversely affected muscle strength. The muscle strength measurements, including PFS and KFS, were normalized to each participant’s body weight to ensure comparability across individuals. All outcome measures were reassessed immediately after the intervention. A blinded researcher performed all the measurements.

### 2.6. Statistical Analysis

To assess the carryover effect, the baseline data between the two periods were compared using an unpaired t-test. Additionally, an analysis of variance (ANOVA) model was constructed with the period and treatment as fixed factors to evaluate the period × treatment interaction. To accurately evaluate the sequence effect, a three-way ANOVA was performed with period, treatment, and group as fixed factors, and the main effect of the sequence was assessed.

To determine whether the CI of the elastic tape applied to the plantar surface and NMES applied to the gastrocnemius muscles produced an effect on DFA equivalent to that of SS, the 95% confidence interval for the difference between the two interventions was calculated. A meta-analysis of SS interventions lasting 5–15 min targeting the triceps surae, as reported in a previous study [4], indicated a mean change of 2.07 (95% confidence interval: 0.86–3.27). On the basis of this meta-analysis, the equivalence margin (δ) was defined as ±1.21 [16]. The utilization of the 95% confidence interval width from meta-analyses appropriately reflects the variability of standard treatments while avoiding the establishment of excessively broad margins. Half the width of the confidence interval derived from the meta-analysis used in this study (±1.21 in the present study) is considered a difference that does not compromise clinical significance [17,18].

The following hypotheses were tested to assess equivalence:

**Null** **Hypothesis** **(H_0_):**
*The mean difference between the CI and SS interventions lies outside the equivalence margin (|M_CI_ − M_SS_| ≥ δ).*


**Alternative** **Hypothesis** **(H_1_):**
*The mean difference between the CI and SS interventions lies within the equivalence margin (|M_CI_ − M_SS_| < δ).*


This approach ensured a rigorous statistical evaluation of the equivalence of the CI and SS.

All statistical analyses were performed using SPSS Statistics (version 27; IBM Corp., Armonk, NY, USA), with a significance level of *p* < 0.05.

## 3. Results

Screening evaluations were conducted in 84 individuals. One participant was excluded because he did not meet the inclusion criteria because he had sustained a knee ligament injury within the past month. Consequently, 83 participants were randomly assigned to two groups (Figure 1). Table 1 presents the participants’ demographic data including age, sex, height, weight, and body mass index. All participants received the assigned interventions and participated in the post-intervention outcome measurements.

To assess the carryover effect, the baseline data between Groups A and B were compared, and the results are presented in Table 2. No significant differences were observed between periods for any of the baseline variables (*p* > 0.05). Additionally, to evaluate the interaction between intervention and period as well as the main effects of intervention and period, an ANOVA model was constructed. The results are summarized in Table 3. No significant interaction effects between the period and intervention were observed for any of the variables (all *p* > 0.05). Similarly, neither the main effect of the period nor the intervention reached statistical significance across all measured outcomes. The effects of intervention (CI, SS), order (CI → SS, SS → CI), and period (first period, second period) were evaluated using a three-way ANOVA. Regarding the order effect, no significant results were observed for main effect of order, intervention × order, intervention × period, or order × period interaction. Specifically, the order effect was not statistically significant, as all *p*-values for the order exceeded 0.05, indicating no significant main effect (DFA, 0.448; FFD, 0.173; SLR, 0.568; PFS, 0.284; KFS, 0.784).

DFA, FFD, and SLR showed significant improvements after both interventions (Table 4). The results of the non-inferiority and equivalence test for DFA indicated that the 95% confidence interval for the difference in DFA between the two interventions fell within the equivalence margin (mean difference: 0.22, 95% confidence interval: −0.72–1.15, equivalence margin: −1.21–1.21) (Figure 2). A linear mixed-effects model was used to compare the changes in DFA between the two interventions, adjusting for period, sequence, and baseline DFA data. The 95% confidence interval for the difference in change scores was −0.822–1.027, which fell within a predefined equivalence margin.

## 4. Discussion

This study is the first randomized crossover trial to demonstrate that a 1-min CI of NMES and elastic taping provides flexibility improvements equivalent to a 5-min SS session in university students. Our findings suggest that the CI of NMES and elastic taping is an effective method for improving flexibility safely and efficiently in a short period.

The CI improved flexibility similarly to SS, surpassing the minimal detectable threshold (MDC95 = 1.4°), highlighting its practical relevance in training and rehabilitation [14].

The observed change in the current study exceeded this threshold, suggesting that the improvement achieved by the CI was not only statistically significant but also clinically meaningful.

SS, traditionally recognized as a representative method for improving flexibility, has been supported by several meta-analyses in terms of its effects on enhancing ROM [6,16]. The frequency and duration of SS may significantly influence the outcomes of flexibility interventions. Previous studies have suggested a dose-response relationship, where longer or more frequent stretching sessions result in greater improvements in ROM. In this study, SS was performed for 5 min, which is a commonly recommended duration; however, it is possible that this dose may not have been sufficient to provide more clinically meaningful results when compared to the CI. Future research should explore the dose-response relationship between SS and the CI to determine whether adjustments in SS dosing could yield greater clinical benefits. Furthermore, the potential trade-offs between increased stretching duration and risks such as muscle strength reduction should also be investigated. However, it has been pointed out that SS requires long intervention times and carries the risk of muscle strength reduction when used before exercise [19,20]. NMES and elastic taping require specialized equipment and expertise, while SS can be performed easily without such resources. However, in sports and rehabilitation settings, interventions that achieve results in a short time are highly valued for their time efficiency. Stretching generally requires a duration of 5 min or more to be effective, whereas the CI of NMES and elastic taping, as demonstrated in this study, can achieve comparable results in just 1 min. This makes the CI particularly suitable for warm-ups or busy clinical schedules. Additionally, SS tends to produce variable results due to individual differences in implementation, such as angle and force. In contrast, NMES and elastic taping can be applied consistently, ensuring high reproducibility. Statistical analyses were conducted to assess the adequacy of the 1-week washout period. The results of ANOVA indicate that the washout period was sufficient to minimize residual effects between the two intervention periods. This further supports the validity and reliability of the randomized crossover design employed in this study. The absence of a significant period and order effects underscores the robustness of the effectiveness and equivalence of the intervention. Moreover, the use of a linear mixed-effects model to adjust for potential confounding factors such as period, sequence, and baseline values adds rigor to the analysis. The consistent finding that the change in DFA between the two interventions fell within the equivalence margin further substantiates the reliability and validity of the equivalence results. By demonstrating consistent results across different periods and sequences, this study provides strong statistical evidence supporting the equivalence and effectiveness of combined NMES and elastic tape intervention compared to SS.

NMES-based interventions are widely used for muscle strength improvement and rehabilitation [21], and several reports have highlighted their effects on flexibility improvement [22,23]. In contrast, elastic tape increases the ROM in distant body parts [10]. Richard et al. [24] quantified skin movement during joint motion and clarified the relationship between joint movement and skin flexibility. We believe that by promoting skin movement with the use of elastic tape, the ROM of distant body parts can be increased. By combining these two methods, this study provides important insights that contribute to the advancement of existing research, suggesting that the CI may be more efficient and effective in improving flexibility than traditional single interventions.

The contribution of NMES and elastic tape to flexibility improvement may be attributed to the enhancement of fascial mechanisms. Fascial mechanisms refer to the role of fascia in transmitting tension and facilitating gliding between tissues. Improvement in fascial glide influences distant body parts through interconnected fascial pathways [25,26]. The superficial back line is a fascial network that connects the soles of the feet to the skull [27], and it is possible that NMES and elastic tape can improve flexibility in distant areas. Applying elastic tape to the plantar fascia creates tension in the skin and fascia, which improves fascial glide [28]. Additionally, NMES improves the gliding of soft tissues, such as the skin and fascia, by forcibly inducing muscle contraction [29]. In this study, FFD and SLR showed improvements similar to those of DFA, suggesting that remote effects through fascial connections may be involved. Previous studies have shown [30,31,32] that fascial connections transmit tension between adjacent structures, indicating that some muscle tension can affect flexibility in other areas. Based on these mechanisms, the intervention method used in this study may have potential applications for improving the overall flexibility of the body.

Safety is a critical consideration in interventions aimed at improving flexibility. This study’s results suggest that NMES, elastic tape, and SS did not cause any significant changes in the muscle strength indicators (PFS and KFS), indicating that these methods are safe and do not lead to side effects, such as muscle weakness. In particular, compared with intervention methods in previous reports [20] suggesting that long-duration SS can cause muscle strength reduction, those in the present study are advantageous because they provide effective results in a short time without any side effects.

This study has several limitations. First, because the participants were limited to healthy university students, the external validity for athletes, older individuals, or patients requiring rehabilitation was limited. To address this issue, further research is required to examine its effects on different populations. This study was conducted as a randomized crossover trial, focusing on the short-term effects of the interventions. While the randomized crossover design provides robust evidence for the immediate equivalence of the CI and SS, the lack of follow-up to assess long-term clinical benefits and practicality is a secondary limitation. Longitudinal studies are needed to evaluate whether the observed short-term effects translate into sustained improvements in flexibility and overall functional outcomes.

## 5. Conclusions

This is the first study to demonstrate that the CI of NMES and elastic tape is an effective and safe method for improving flexibility over a short period. This suggests its potential as an efficient alternative to SS in competitive sports and health-promotion programs. The CI of NMES and elastic tape, which improves flexibility within a short period, could be particularly advantageous in scenarios such as brief warm-ups before matches or time-constrained rehabilitation settings. Future studies are needed to validate the practicality of this intervention by expanding the target population to athletes, patients, and older individuals while assessing its long-term effects over periods ranging from a few hours to several months. This research could also provide deeper insight into its broader applications and sustainability across different health conditions.

## Figures and Tables

**Figure 1 jfmk-10-00058-f001:**
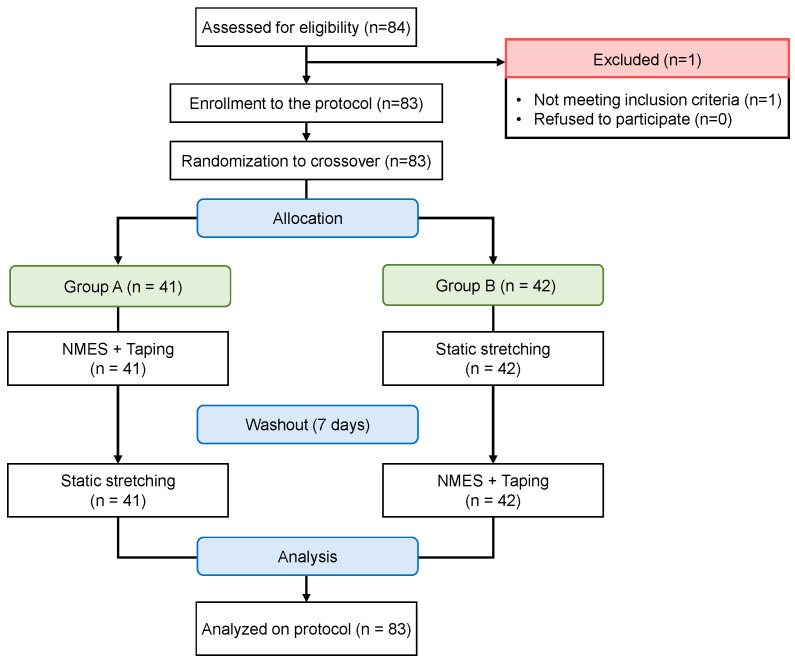
Flowchart of the crossover process. NMES, neuromuscular electrical stimulation.

**Figure 2 jfmk-10-00058-f002:**
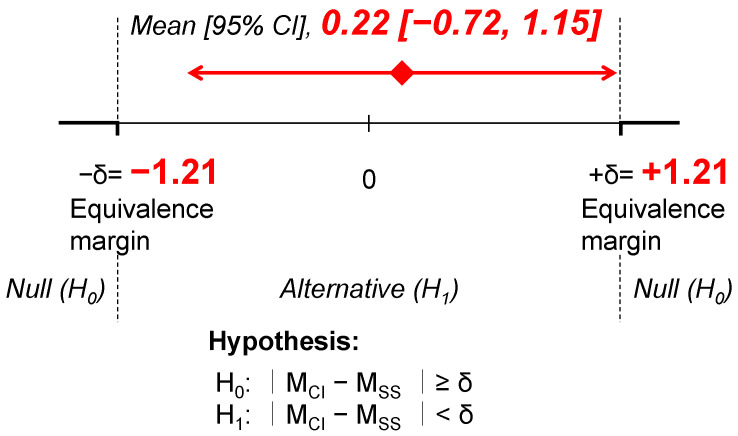
Difference in DFA from baseline to each follow-up between CI and SS. M = (DFA immediately after intervention) − (DFA at baseline). M_CI_, combined intervention; M_SS_, static stretching; CI, confidence interval.

**Table 1 jfmk-10-00058-t001:** Participants’ demographic data.

	Total (*n* = 83)	Group A (*n* = 41) CI First	Group B (*n* = 42) SS First
Age, years	20.7 ± 0.9	20.6 ± 0.9	20.9 ± 0.8
Sex, *n* (%)			
Male	42 (51)	21 (51.2)	21 (50)
Female	41 (49)	20 (48.8)	21 (50)
Height, cm	165.1 ± 7.9	165.2 ± 7.7	165.1 ± 8.2
Weight, kg	59.4 ± 11.4	60.0 ± 10.7	58.9 ± 12.2
BMI, kg/m^2^	21.6 ± 2.9	21.9 ± 3.0	21.4 ± 2.9

Data are presented as mean ± standard deviation. BMI, body mass index; CI, combined intervention; SS, static stretching.

**Table 2 jfmk-10-00058-t002:** Comparison of baseline data between periods.

	Group A	Group B	Mean Difference	95% Confidence Interval		*p*-Value
				Lower	Upper	
DFA, °	39.6 ± 6.3	41.7 ± 6.3	−2.1	−4.8	0.7	0.143
FFD, cm	2.2 ± 9.4	4.1 ± 8.4	−1.9	−5.7	2.0	0.345
SLR, °	68.0 ± 15.4	69.0 ± 14.4	−1.0	−7.5	5.5	0.754
PFS, Nm/kg	1.61 ± 0.42	1.61 ± 0.34	0.00	−0.16	0.17	0.962
KFS, Nm/kg	1.26 ± 0.32	1.88 ± 4.30	−0.63	−1.96	0.71	0.355

Data are presented as mean ± standard deviation. DFA, dorsiflexion angle; FFD, finger-floor distance; SLR, straight leg raising; PFS, plantar flexor strength; KFS, knee flexor strength.

**Table 3 jfmk-10-00058-t003:** Evaluation of carryover effects in the interaction between intervention and period.

		Mean Square	F	*p*-Value
DFA	Period × Intervention	18.09	0.579	0.448
	Period	0.24	0.008	0.931
	Intervention	22.36	0.715	0.399
FFD	Period × Intervention	134.72	1.871	0.173
	Period	1.05	0.015	0.904
	Intervention	0.02	0.000	0.988
SLR	Period × Intervention	66.29	0.327	0.568
	Period	0.00	0.000	0.999
	Intervention	0.60	0.003	0.957
PFS	Period × Intervention	1.81	1.155	0.284
	Period	1.26	0.803	0.371
	Intervention	0.71	0.455	0.501
KFS	Period × Intervention	0.01	0.076	0.784
	Period	0.01	0.130	0.718
	Intervention	0.14	1.464	0.228

DFA, dorsiflexion angle; FFD, finger-floor distance; SLR, straight leg raising; PFS, plantar flexor strength; KFS, knee flexor strength.

**Table 4 jfmk-10-00058-t004:** Comparison of clinical outcomes after the intervention.

	Post-Intervention		Mean Change from Baseline				CI-SS (95% Confidence Interval)
	CI	SS	CI (95% Confidence Interval)	*p*-Value	SS (95% Confidence Interval)	*p*-Value	
DFA, °	38.5 ± 5.4	37.7 ± 5.7	−2.8 (−3.4 to −2.1)	<0.001	−2.6 (−3.2 to −1.9)	<0.001	0.21 (−0.72 to 1.15)
FFD, cm	4.7 ± 8.3	4.2 ± 8.7	2.0 (1.5 to 2.4)	<0.001	2.1 (1.7 to 2.5)	<0.001	0.1 (−0.5 to 0.6)
SLR, °	70.6 ± 14.2	70.5 ± 14.2	2.6 (1.5 to 3.6)	<0.001	3.1 (1.9 to 4.3)	<0.001	0.6 (−0.9 to 2.0)
PFS, Nm/kg	1.66 ± 0.43	1.79 ± 1.72	0.07 (−0.00 to 0.14)	0.065	0.16 (−0.22 to 0.53)	0.407	0.09 (−0.30 to 0.47)
KFS, Nm/kg	1.26 ± 0.30	1.20 ± 0.31	−0.38 (−1.09 to 0.32)	0.279	−0.03 (−0.07 to 0.01)	0.091	0.35 (−0.35 to 1.06)

Data are presented as mean ± standard deviation. DFA, dorsiflexion angle; FFD, finger-floor distance; SLR, straight leg raising; PFS, plantar flexor strength; KFS, knee flexor strength; CI, combined intervention; SS, static stretching.

## Data Availability

Data is contained within the article.

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
