# Peer review of "Combined Neuromuscular Electrical Stimulation and Elastic Taping Improves Ankle Range of Motion Equivalent to Static Stretching in Untrained Subjects"

_jfmk, 2025, doi:10.3390/jfmk10010058_

Round 1

Reviewer 1 Report

Comments and Suggestions for Authors

The study aimed to evaluate the effects of combining neuromuscular electrical stimulation (NMES) and elastic tape in improving ankle joint range of motion (ROM) compared to traditional static stretching (SS). The results indicate that both SS and NMES/ taping improved ROM equally. The manuscript is well written and statistical analysis robust.

It is suggested the authors make a number of adjustments and reply before the manuscript can be considered for publications.

The title should be modified so that a re informative title is used. ‘Combined NMES and taping improves ankle ROM equal to static stretching in untrained subjects’. Further to this recommendation, the authors should generally advocate (especially in the discussion)that NMES/ taping Improves ROM  it discuss the practical aspects of administration of NMES/ taping relative to stretching which requires no equipment or clinically expertise.

In terms of methodology , it is claimed this study is  a randomized crossover trial- was it truely a RCT? There was no follow up- therefore the clinical benefits or practicality is unclear.  

The study was conducted in 83 healthy university students. Would it have been better to conduct analysis based on baseline ROM? Please also provide an analysis of data based on high and low responders or whether ROM at baseline was a co- variant.

Static stretching frequency and length of needs clarification? Dose - response relationship is not explored between control and intervention. Hence the SS may not have been sufficient dose to provide more clinical meaningful results. Please discuss this in the discussion section.

Why was a title table used for static stretching ? This equipment is not widely available. There is also many other SS exercises that court have been used. 

Was a 1-week washout period- long enough?

The outcome measurements included  dorsiflexion angle (DFA), finger-floor distance (FFD), straight leg raise (SLR), and muscle strength (plantar flexor strength and knee flexor strength),- why strength? There was no effects on this measure.  Remove? Why SLR when intervention was on the ankle ?

Discussion should include dose and repose of stretching and whether a greater dose may have been more clinically beneficial .

Specific comments

Line 67 ‘Additionally, NMES alleviates pain and elevates pain thresholds [8,9], indicating its potential role in improving ROM. ‘ Why would alleviation of pain improve ROM? Discuss 

What is fascial mechanisms? Provide further informative 

Line 78 not sure ‘efficacy’ is the right word. Replace.

Author Response

Reviewer1

The study aimed to evaluate the effects of combining neuromuscular electrical stimulation (NMES) and elastic tape in improving ankle joint range of motion (ROM) compared to traditional static stretching (SS). The results indicate that both SS and NMES/ taping improved ROM equally. The manuscript is well written and statistical analysis robust. It is suggested the authors make a number of adjustments and reply before the manuscript can be considered for publications.

#1. The title should be modified so that are informative title is used. ‘Combined NMES and taping improves ankle ROM equal to static stretching in untrained subjects’.

Response: Thank you for your suggestion. To better reflect the content of the study, we have revised the title to: "Combined Neuromuscular Electrical Stimulation and Elastic Taping Improves Ankle Range of Motion Equivalent to Static Stretching in Untrained Subjects."

#2. Further to this recommendation, the authors should generally advocate (especially in the discussion) that NMES/ taping Improves ROM  it discuss the practical aspects of administration of NMES/ taping relative to stretching which requires no equipment or clinically expertise.

Response: Thank you for your constructive comment. We have revised the discussion section to explicitly compare the practical aspects of the two interventions. While NMES and taping require specialized equipment and expertise, stretching can be performed easily without such resources. However, in sports and rehabilitation settings, interventions that achieve results in a short time are highly valued for their time efficiency. Stretching generally requires a duration of 5 minutes or more to be effective, whereas, as demonstrated in this study, the combined intervention of NMES and elastic taping can achieve comparable results in just one minute. This makes the combined intervention particularly suitable for warm-ups or busy clinical schedules. Additionally, static stretching tends to produce variable results due to individual differences in implementation, such as angle and force. In contrast, NMES and taping can be applied consistently, ensuring high reproducibility (Line 285-294).

#3. In terms of methodology , it is claimed this study is  a randomized crossover trial- was it truely a RCT? There was no follow up- therefore the clinical benefits or practicality is unclear.  

Response: This study was indeed a randomized crossover trial, focusing on the short-term effects of the interventions. We acknowledge the lack of follow-up as a limitation and have explicitly noted this in the manuscript. Additionally, we have included the necessity to investigate long-term effects in the conclusion (Line 356-362).

#4. The study was conducted in 83 healthy university students. Would it have been better to conduct analysis based on baseline ROM? Please also provide an analysis of data based on high and low responders or whether ROM at baseline was a co- variant.

Response: Thank you for your insightful comment. In this study, we conducted analyses using a linear mixed-effects model, including baseline ROM, period, and sequence as covariates. The results indicated that even after adjustment, the combined intervention of NMES and elastic taping demonstrated equivalent ROM improvement effects compared to static stretching. These findings are detailed in the Methods and Results sections of the manuscript. Additionally, the analysis of data based on high and low responders (responders and non-responders) is an important consideration for future research. Such analyses could provide valuable insights into the potential for individualized interventions.

#5. Static stretching frequency and length of needs clarification? Dose - response relationship is not explored between control and intervention. Hence the SS may not have been sufficient dose to provide more clinical meaningful results. Please discuss this in the discussion section.

Response: Thank you for your valuable comment. We have added a discussion on the potential influence of static stretching frequency and duration on the outcomes. Additionally, we have highlighted the need for future research to explore the dose-response relationship between static stretching and the combined intervention, as insufficient dosing of static stretching may have limited its clinical impact in this study (Line 274-283).

#6. Why was a title table used for static stretching ? This equipment is not widely available. There is also many other SS exercises that court have been used.

Response: We used a tilt table with a fixed backrest to maintain a stable stretching setup. The footplate angle was individually adjusted to each participant’s maximum tolerable stretch, and this angle was kept constant throughout the 5-minute intervention. Although the absolute angles varied among participants, each individual received a consistent stretching intensity, thereby minimizing within-participant variability in the stretching protocol (Line 140-148).

#7. Was a 1-week washout period- long enough?

Response: Thank you for your comment. In this study, a 1-week washout period was implemented. Statistical analyses were conducted to assess period effects and carryover effects, and the period × intervention interaction was found to be non-significant. Based on these results, we concluded that a 1-week washout period was sufficient. This point was already addressed in the Methods and Discussion sections of the manuscript. In the revised version, we have further emphasized in the Discussion section that the absence of carryover and period effects supports the adequacy of the washout period (Line 299-304).

#8. The outcome measurements included  dorsiflexion angle (DFA), finger-floor distance (FFD), straight leg raise (SLR), and muscle strength (plantar flexor strength and knee flexor strength),- why strength? There was no effects on this measure.  Remove? Why SLR when intervention was on the ankle ?

Response: We have clarified in the Methods section that muscle strength measurements were included to evaluate the safety of the interventions, ensuring that no negative impact occurred on strength. Additionally, SLR was included to assess overall changes in range of motion throughout the body, beyond the specific focus on the ankle (Line 151-153,166-168).

#9. Discussion should include dose and repose of stretching and whether a greater dose may have been more clinically beneficial .

Response: Thank you for your valuable comment. We have added a discussion on the potential influence of static stretching frequency and duration on the outcomes. Additionally, we have highlighted the need for future research to explore the dose-response relationship between static stretching and the combined intervention, as insufficient dosing of static stretching may have limited its clinical impact in this study. Furthermore, we have also discussed the possibility that increasing the dose of stretching might provide greater clinical benefits, while noting the potential risk of adverse effects, such as muscle strength reduction, associated with excessive stretching (Line 274-283).

#10. Line 67 ‘Additionally, NMES alleviates pain and elevates pain thresholds [8,9], indicating its potential role in improving ROM. ‘ Why would alleviation of pain improve ROM? Discuss 

Response: Thank you for your comment. The potential mechanisms by which alleviation of pain improves ROM include the reduction of muscle tension that restricts joint mobility, allowing for smoother movement. Additionally, pain relief can decrease avoidance behaviors related to movement, enabling individuals to fully utilize their range of motion. This explanation has been incorporated into the intruduction section of the manuscript (Line 67-69).

#11. What is fascial mechanisms? Provide further informative 

Response: Thank you for your comment. Fascial mechanisms refer to the role of fascia in transmitting tension and facilitating gliding between tissues. The application of elastic tape is thought to enhance fascial glide, which may contribute to improvements in range of motion (ROM). This effect is believed to result from the redistribution and equalization of tension across the fascial network. We have added this explanation to the Methods and Discussion sections of the manuscript for further clarification (Line 69-75,319-325).

#12. Line 78 not sure ‘efficacy’ is the right word. Replace.

Response: Thank you for your comment. Upon review, we agree that "efficacy" may not be the most appropriate term in the context of this study. We have replaced it with "effectiveness," which better reflects the study's focus.

Reviewer 2 Report

Comments and Suggestions for Authors

Effect of Combined Neuromuscular Electrical Stimulation and Elastic Tape on Ankle Joint Range of Motion: A Randomized  Crossover Trial

Interesting and with practical benefits work. The manuscript in my opinion is valuable and significant. The manuscript structure is good and it is obvious that the Authors have done an enough work. Personally, I learned important information. I propose some notes to improve the manuscript quality.

1. Introduction

Тhis paragraph is well-formed in terms of meaning, but it is not in terms of depth. For me, it is necessary to analyze more literary sources in terms of the same approaches and their results presented from other author teams. Why the authors were decided that the elastic tape intervention will improve ankle ROM?

The sentence (69-71) – “Notably, applying elastic tape to the sole of the foot improves ankle ROM via fascial mechanisms, making it particularly suitable for short-term applications before athletic performance” must be added and continued!

2. Materials and Methods

The sample of people included in the study is enough.

3. Results

In my opinion this section is excellent presented in accordance to the obtained data structure.

4. Discussion

This work has its limitations which are presented here.

The Discussion is laid out consistently and clearly.

5. Conclusion.

A well-formed conclusion. Authors do well to include some specifically list with the benefits for coaches or/and physiotherapists of their research.

6. Abstract

When the authors revise the Conclusion the Abstract can be rewritten. The unwritten rule is that most readers only look at these paragraphs – abstract and conclusion.

Author Response

Reviewer 2

Effect of Combined Neuromuscular Electrical Stimulation and Elastic Tape on Ankle Joint Range of Motion: A Randomized  Crossover Trial. Interesting and with practical benefits work. The manuscript in my opinion is valuable and significant. The manuscript structure is good and it is obvious that the Authors have done an enough work. Personally, I learned important information. I propose some notes to improve the manuscript quality.

#1. Introduction

Тhis paragraph is well-formed in terms of meaning, but it is not in terms of depth. For me, it is necessary to analyze more literary sources in terms of the same approaches and their results presented from other author teams. Why the authors were decided that the elastic tape intervention will improve ankle ROM? The sentence (69-71) – “Notably, applying elastic tape to the sole of the foot improves ankle ROM via fascial mechanisms, making it particularly suitable for short-term applications before athletic performance” must be added and continued!

Response: Thank you for your suggestion. We have revised the Introduction section to include additional references to studies conducted by other research teams, highlighting the potential of elastic tape interventions to improve ankle ROM through fascial mechanisms. These additions provide further context and support for our statement (Line 73-75).

#2. Materials and Methods

The sample of people included in the study is enough.

Response: Thank you for your positive comment regarding the sample size.

#3. Results

In my opinion this section is excellent presented in accordance to the obtained data structure.

Response: Thank you for your kind feedback. We are pleased to hear that you found the Results section to be well-presented and aligned with the data structure.

#4. Discussion

This work has its limitations which are presented here.

The Discussion is laid out consistently and clearly.

Response: Thank you for your thoughtful feedback. We appreciate your recognition of the clarity and consistency in the Discussion section, as well as the acknowledgment of how the study’s limitations have been presented.

#5. Conclusion.

A well-formed conclusion. Authors do well to include some specifically list with the benefits for coaches or/and physiotherapists of their research.

Response: Thank you for your valuable suggestion. We have revised the conclusion to explicitly highlight how coaches and physiotherapists can utilize NMES and elastic tape to improve flexibility in a short time. Additionally, we emphasized the practical benefits of these interventions for warm-ups and rehabilitation (Line 369-372).

#6. Abstract

When the authors revise the Conclusion the Abstract can be rewritten. The unwritten rule is that most readers only look at these paragraphs – abstract and conclusion.

Response: Thank you for your feedback. In line with the revisions made to the Conclusion section, we have updated the Abstract to concisely reflect the significance of the study and its practical benefits. This ensures consistency and highlights key takeaways for readers (Line 38-42).